# Antagonism of Protein Kinase R by Large DNA Viruses

**DOI:** 10.3390/pathogens11070790

**Published:** 2022-07-12

**Authors:** Annabel T. Olson, Stephanie J. Child, Adam P. Geballe

**Affiliations:** 1Divisions of Human Biology and Clinical Research, Fred Hutchinson Cancer Center, 1100 Fairview Ave N Seattle, P.O. Box 19024, Seattle, WA 98109, USA; atolson@fredhutch.org (A.T.O.); schild@fredhutch.org (S.J.C.); 2Departments of Microbiology, University of Washington, Seattle, WA 98195, USA; 3Departments of Medicine, University of Washington, Seattle, WA 98195, USA

**Keywords:** vaccinia virus, cytomegalovirus, E3L, K3L, TRS1, protein kinase R, dsRNA, RNaseL, evolutionary arms race, gene amplification

## Abstract

Decades of research on vaccinia virus (VACV) have provided a wealth of insights and tools that have proven to be invaluable in a broad range of studies of molecular virology and pathogenesis. Among the challenges that viruses face are intrinsic host cellular defenses, such as the protein kinase R pathway, which shuts off protein synthesis in response to the dsRNA that accumulates during replication of many viruses. Activation of PKR results in phosphorylation of the α subunit of eukaryotic initiation factor 2 (eIF2α), inhibition of protein synthesis, and limited viral replication. VACV encodes two well-characterized antagonists, E3L and K3L, that can block the PKR pathway and thus enable the virus to replicate efficiently. The use of VACV with a deletion of the dominant factor, E3L, enabled the initial identification of PKR antagonists encoded by human cytomegalovirus (HCMV), a prevalent and medically important virus. Understanding the molecular mechanisms of E3L and K3L function facilitated the dissection of the domains, species-specificity, and evolutionary potential of PKR antagonists encoded by human and nonhuman CMVs. While remaining cognizant of the substantial differences in the molecular virology and replication strategies of VACV and CMVs, this review illustrates how VACV can provide a valuable guide for the study of other experimentally less tractable viruses.

## 1. Introduction

The study of poxviruses, particularly vaccinia virus (VACV), has revealed numerous intricate molecular interactions with the host cell that enable the virus to replicate and produce infectious progeny [1]. Because many viruses face similar challenges, including the evasion of myriad intrinsic host defense systems, understanding how VACV overcomes these systems can provide a useful conceptual framework for studying other viruses. VACV replicates rapidly, has a broad host range, and is relatively easy to genetically engineer. Thus, it also provides a tractable experimental system for generating recombinant viruses to analyze cellular interactions with proteins from other viruses, including those that are more challenging to study. Following its successful use for the worldwide eradication of smallpox, VACV research has continued to enrich our understanding of the fundamental molecular virology, immunology, and evolution of poxviruses. VACV recombinants have been used in applications such as the development of vaccines and oncolytic agents, expression of cDNA libraries, surface display of antigens, antibody libraries, and even the propagation and genetic manipulation of RNA viruses, including coronaviruses [2,3,4,5,6,7,8,9].

Our lab has taken advantage of the power of the VACV system to study the medically important pathogen human cytomegalovirus (HCMV). HCMV infects an estimated 50–90% of humans worldwide [10]. Though most HCMV infections cause few or no symptoms, the virus does cause serious and sometimes fatal disease in patients with poor immune system function. HCMV is also a major cause of congenital diseases, including microcephaly, developmental delay, and hearing and vision loss. After the initial infection subsides, HCMV persists in a latent state, likely for the rest of an individual’s life, with the potential to reactivate if immune system function declines. Moreover, life-long latent HCMV infection appears to exert subtle but pervasive detrimental effects on the immune system, with potentially broad consequences for aging and human health [11].

Using VACV to study HCMV takes advantage of similarities but also requires consideration of important differences between these viruses [1,10]. Like VACV, HCMV has a large dsDNA genome (~190 and ~230 kb, respectively), which is packaged along with viral proteins inside an enveloped viral particle that is from ~200 to 300 nm in diameter. Both viruses encode over 200 genes that are expressed in a temporally regulated manner. Like many viruses, infection with VACV leads to the shutoff of host cell protein synthesis, while infection with HCMV does not. Unlike most DNA viruses including HCMV, VACV replicates in the cytoplasm and relies on its own RNA polymerase for viral gene transcription. Lastly, HCMV replicates more slowly than VACV and has a limited host range. Because of these and other differences, insights emerging from studies of HCMV genes using VACV experimental systems generally require confirmatory studies in the more natural HCMV context.

We have primarily used VACV to study the interactions of HCMV and related nonhuman CMVs with the broadly acting host defense pathway mediated by protein kinase R (PKR). The detection of double-stranded RNA (dsRNA) by PKR triggers the shutoff of protein synthesis, thereby limiting viral replication [12]. Although its source is uncertain, dsRNA accumulates even during DNA virus infections [13,14,15]. VACV and HCMV each produce transcripts from both genomic strands, some of which have complementary sequences with the potential to anneal and form dsRNA. Although evolutionary pressure to maintain a compact genome may be an “Achilles heel” for viruses as it increases the potential for production of dsRNA, many viruses have evolved effective countermeasures that inhibit one or more steps in the PKR pathway and thereby enable continued protein synthesis and viral replication [16].

## 2. Poxviral Antagonists of Host Defense Pathways

Among its large repertoire of genes, VACV encodes two PKR antagonists, E3L and K3L. Identification of these factors followed the early discovery that VACV was relatively resistant to interferon (IFN) and that co-infection with VACV can rescue the replication of IFN-sensitive viruses [17,18,19]. A pivotal discovery in understanding VACV’s IFN resistance was the finding that E3L is a potent antagonist of PKR [20,21]. VACV from which E3L is deleted (VACV∆E3L) cannot replicate efficiently in human cells or in many other mammalian cell lines, but it does replicate in PKR-deficient derivatives [22,23,24,25,26]. These observations demonstrate that a critical role of E3L is to counteract the PKR pathway, thereby enabling VACV replication.

E3L also prevents activation of another dsRNA-activated host defense system, the oligoadenylate synthetase (OAS)/RNaseL pathway. If unchecked, RNaseL degrades both host and viral RNAs, thus restricting viral replication [27]. Deletion of RNaseL has very little effect on the lethality of VACV∆E3L in mice, while deletion of PKR increased mortality modestly. Deletion of both RNAseL and PKR had a larger effect, suggesting that E3L blocks both of these host defense systems in this in vivo model [28].

Studies performed during the past two decades have characterized several important structural and functional features of E3L [29]. Due to the use of an alternative in-frame start codon, the *E3L* gene expresses both a 25 kilodalton (kDa) protein and an N-terminally truncated 20 kDa isoform. Both proteins contain a critical C-terminal dsRNA-binding domain (dsRBD, Figure 1) with homology to those found in many other dsRNA-binding proteins [30,31]. This region of E3L also contains a dimerization domain. In the context of cell culture, the C-terminal domain is often sufficient to impede, but not eliminate, activation of the PKR and OAS/RNaseL pathways and to allow viral replication [32,33].

Because the N-terminal domain of E3L was found to be dispensable for viral replication in cell culture experiments, it was surprising to find that it is necessary for VACV virulence in mice [34]. Subsequent studies showed that this domain contains a Z-form nucleic acid (Z-NA) binding domain (ZBD, Figure 1) and, notably, replacement of this ZBD with homologous domains from other proteins restores virulence in mice [35]. The ZBDs of ADAR1, which acts to repress the interferon response to dsRNA and Z-NA [36], Z-NA-binding protein 1 (ZBP-1, also called DAI), and E3L are functionally interchangeable in this system [35]. Thus, the contribution of the E3L N-terminus to virulence in mice appears to depend at least in part on Z-NA binding.

Binding and sequestration of Z-NA by the E3L ZBD prevents the activation of ZBP1-RIPK3-MLKL-driven necroptosis [36]. This was only recently uncovered because this mechanism requires an intact necroptosis pathway, which is present in mice but lacking in many of the cell lines commonly used for studying VACV. Remarkably, Koehler et al. reported that dsRNA binding by E3L induces the formation of Z-form RNA. When the N-terminal ZBD is missing, the induced Z-RNA is free to bind to ZBP-1, triggering necroptosis and inhibiting viral replication [36]. The N-terminal E3L deletion mutant virus is avirulent in wild-type and PKR-null mice, while replication is restored in RIP3-null or ZBP-1-null mice [36]. In contrast, VACV∆E3L is not pathogenic in wild-type mice, but infectivity is partially restored in PKR-null mice [28]. VACV∆E3L does not activate necroptosis, supporting the conclusion that E3L lacking its N-terminal ZBD triggers necroptosis as a result of the C-terminus in inducing formation of ZBP-1-activating Z-RNA [37].

To further complicate matters, a few studies have shown that the N-terminal domain of E3L contributes to binding and antagonism of PKR [33,38,39]. Thakur et al. identified point mutants in the N-terminal domain of variola virus E3L that impaired Z-NA binding but still retained the N-terminal region’s function of inhibiting PKR in yeast and in dsRNA-treated mammalian cells. Conversely, they identified two mutants that bound to both dsRNA and Z-NA but were unable to antagonize PKR, demonstrating a role for the N-terminus in Z-NA-independent inhibition of PKR. Thus, studies of E3L continue to reveal new complexity in the interactions of dsRNA and Z-NA biology and their roles in disease pathogenesis.

**Figure 1 pathogens-11-00790-f001:**
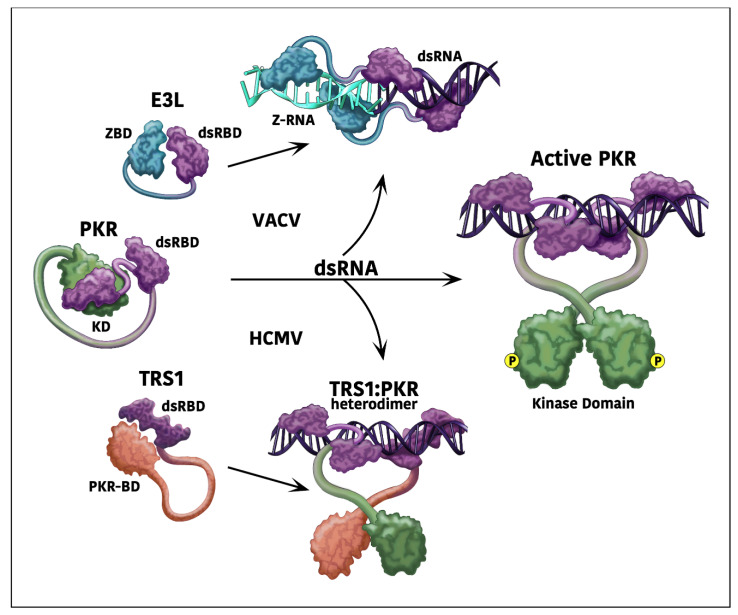
**Model of PKR antagonism by E3L and TRS1.** When dsRNA is produced during viral infections, PKR undergoes a conformational change, leading to activation by homodimerization and autophosphorylation [12,40]. VACV E3L can bind to and sequester dsRNA, thereby preventing PKR activation [29,40]. Binding of E3L to dsRNA induces the formation of Z-RNA, which can trigger necroptosis. However, the N-terminal domain of E3L binds to this Z-RNA, blocking activation of the necroptosis signaling pathway. HCMV TRS1 binds to both dsRNA and PKR, possibly forming a heterodimer that prevents PKR activation [23,41]. The dsRBDs of each protein are shown in purple.

The discovery that VACV K3L could also antagonize PKR was established following the observation that VACV∆E3L retained interferon resistance in some contexts [42]. K3L is ~30% identical to eIF2α, the key substrate of PKR, and analyses suggested that K3L was likely acquired by horizontal gene transfer of the host gene encoding eIF2α. [43]. Interestingly, VACV lacking K3L (VACV∆K3L) replicates efficiently in many cell types, except for some IFN-treated rodent cells as well as gibbon fibroblasts [42,43,44]. These results suggest that K3L exhibits species-specificity in antagonizing PKR. In functional assays in yeast and mammalian cells, K3L only weakly antagonizes human PKR but is more effective in counteracting PKR alleles from gibbons, New and Old World monkeys, cows, and rodents [44,45,46]. A few mutations that increase the ability of K3L to antagonize human PKR have been identified. For example, mutation of the histidine at codon 47 to arginine (H47R) enhances K3L-mediated inhibition of human PKR. This mutant was identified in two independent screens, one using random mutagenesis in a yeast assay and the other by serial passage of VACV∆E3L in human cells [47,48]. K3L was also reported to increase VACV dissemination from the lung to secondary sites in mice, although whether this phenotype is linked to antagonism of PKR by K3L is not known [28].

With its wide host range, fast replication rate, and relative IFN resistance, VACV offers an attractive model system for studying IFN antagonists from other viruses that are more difficult to study. VACV∆E3L has been used as a model for studying PKR antagonists from diverse viruses, including the Orf virus E3L homolog, porcine group C rotavirus NSP3, Influenza NS1, and the MHV nucleocapsid protein, as well as the dsRNA-binding *E. coli* RNase III [27,49,50,51,52,53]. In addition, our lab has used the VACV∆E3L system to identify and study PKR antagonists from several CMV species, as described next.

## 3. Complementation of VACV∆E3L by HCMV

Although we did not initially have any compelling reason to suspect that HCMV might encode one or more PKR antagonists, the presence of such factors in other large DNA viruses and in many RNA viruses led us to investigate this possibility. Because HCMV replicates slowly and does not shut off cellular protein synthesis, we were able to evaluate this question using a sequential infection assay. We found that infection of human fibroblasts (HF) with HCMV followed by infection with VACV∆E3L resulted in almost complete rescue of the 1000-fold replication defect seen with VACV∆E3L in these cells [54]. This experiment provided the first indication that HCMV does indeed have one or more genes that can block PKR. Importantly, HCMV did not increase wild-type VACV titers, demonstrating that the rescue of VACV∆E3L replication by HCMV was specific in compensating for the loss of E3L and not a general stimulatory effect of HCMV on VACV replication. In fact, HCMV inhibited wild-type VACV replication up to 10-fold, which is not too surprising as these are large complex viruses that likely compete for limited cellular resources or encode factors that might interact in functionally disruptive ways. In addition to rescuing VACV∆E3L replication, HCMV blocked eIF2α phosphorylation, OAS/RNaseL activation, and the shutoff of protein synthesis that are caused by VACV∆E3L infection. These results strongly suggested that HCMV has one or more genes that can functionally substitute for E3L.

## 4. Identification of the HCMV PKR and OAS/RNaseL Antagonists Using VACV∆E3L

We further exploited VACV∆E3L to identify the HCMV-encoded antagonists by constructing a library of VACV∆E3L recombinants containing HCMV genomic fragments [55]. After infection of HF with this library, we detected a single plaque, which was notable because VACV∆E3L does not normally form plaques in HF. Sequencing revealed that this virus contained the entire HCMV TRS1 coding region. TRS1 belongs to the CMV US22 gene family, a group of ~12 genes with conserved sequence motifs, suggesting that they arose due to gene duplication [56]. These genes subsequently evolved to serve divergent functions. In all CMVs studied thus far, one or two of the US22 family members serve as PKR antagonists (Figure 2). Because the 5′ 2/3 of the *TRS1* gene is located within the inverted repeats flanking the unique short region of the HCMV genome, *TRS1* is identical to the 5′ 2/3 of a second gene, *IRS1*, that is partially encoded within the other repeat. The remaining 1/3 of these genes is also ~50% conserved at the amino acid level. We constructed VACV∆E3L recombinants expressing either TRS1 or IRS1 and found that both complemented VACV∆E3L replication and prevented the shutoff of translation, eIF2α phosphorylation, and OAS/RNaseL activation.

Prior to our identification of TRS1 and IRS1 as PKR antagonists, several groups had constructed HCMV viruses with mutations of TRS1 or IRS1 [62,63,64,65]. Analyses of these viruses revealed that neither TRS1 nor IRS1 is essential for HCMV replication. Because TRS1 and IRS1 were functionally redundant in overcoming PKR in the context of VACV∆E3L, we hypothesized that they might be redundant in the context of HCMV as well. Indeed, HCMVs lacking both IRS1 and TRS1 are unable to block PKR and, notably, do not replicate at all in wild-type HF, but they do replicate well in PKR-null HF (Table 1, [23,66]). Further support for the conclusion that TRS1 and IRS1 are functionally analogous to E3L came from the finding that insertion of E3L into HCMV∆I/∆T rescued HCMV replication [13]. Moreover, insertion of TRS1 into other viruses with defects in PKR antagonism, such as herpes simplex 1 with a mutation in the γ34.5 gene and mouse CMV (MCMV) lacking the TRS1 homologs m142 and m143 improved the replication of these viruses, consistent with the ability of TRS1 to antagonize PKR in these contexts as well [59,67]. Interestingly, insertion of E3L into MCMV deleted of m142 and m143 can partially rescue replication in cell culture, demonstrating the functional interchangeability of these antagonists [68].

Several reports have suggested that TRS1 and IRS1 might also have roles in other processes such as transcription, translation of specific mRNAs, and inhibition of autophagy [69,70,71,72,73]. Some of these effects might be indirect consequences of the inhibition of PKR. Regardless, the observation that HCMV∆I/∆T replicates as well as wild-type HCMV in PKR-null cells suggests that, at least in these conditions, the only essential function of these genes is evasion of PKR [23,66].

Although VACV∆E3L proved to be an invaluable aid in identifying these HCMV PKR antagonists, we did encounter limitations with this strategy. For example, while IRS1 and TRS1 are functionally indistinguishable in the context of VACV∆E3L, they have substantially different phenotypes in the context of HCMV. Deletion of IRS1 alone has no effect on HCMV replication, while deletion of TRS1 yields a substantial replication defect [62,63,64,65]. One possible explanation for this discrepancy is that IRS1 expresses a truncated protein product from an internal promoter. This smaller protein was shown to inhibit the stimulatory effect of TRS1 or IRS1 on HCMV gene expression [70]. The internal promoter would likely not be recognized by the VACV RNA polymerase, so this potentially inhibitory protein would not be expressed from VACV recombinants that contain the entire IRS1 gene. Thus, differences in transcriptional regulatory mechanisms, among other factors, require consideration when extrapolating conclusions from studies using VACV to the understanding of CMV biology.

## 5. Mechanisms of HCMV Antagonism of dsRNA-Activated Host Defenses

Because the dsRBD of E3L is sufficient to complement VACV∆E3L replication in cell culture [74], we hypothesized that TRS1 and IRS1 might also function by simply binding to and sequestering dsRNA. Although TRS1 and IRS1 lack apparent sequence homology with known dsRNA binding proteins, their identical N-terminal region does bind to dsRNA [75]. However, the dsRBD alone did not allow VACV∆E3L replication. PKR binding by the C-terminus of either TRS1 or IRS1 was also necessary [23,66,76].

We used binding assays to delineate residues in TRS1 needed for direct interactions with dsRNA and with PKR [23,41]. Mutations disrupting either of these functions eliminated the ability of TRS1 to rescue VACV∆E3L replication. Importantly, introduction of either of these TRS1 mutants into HCMV∆I/∆T failed to rescue replication (Figure 3), confirming that the same critical binding properties identified using VACV∆E3L are also essential for HCMV replication (Figure 3, [23,66]).

Expression of either TRS1 or IRS1 prevents activation of the OAS/RNaseL pathway after VACV∆E3L infection [55]. Surprisingly, despite the accumulation of dsRNA in HCMV-infected cells, the OAS/RNaseL pathway remains inactive in HCMV[∆I/∆T]-infected cells [13]. Although TRS1 and IRS1 are not required to block OAS/RNaseL under the conditions used in these experiments, it remains possible that their ability to block this pathway plays a role in other cell types or under different conditions. However, studies in MCMV with deletions of its antagonists (m142 and m143, Figure 2) revealed little if any role for RNaseL antagonism in vivo [68]. Differences in the quantities, timing of expression, or specific features of the dsRNAs may explain this discrepancy in activation of the OAS/RNaseL pathway between the VACV and CMV systems [38].

## 6. Interrogation of the CMV-PKR Evolutionary “Arms Race” Using VACV Recombinants

The variable activity of K3L in different cell types can be explained at least in part by species-specific variations in PKR. Two groups have reported that PKR has been undergoing adaptive evolution for millions of years [44,46]. Mutational analyses guided by computational evaluations of the rapidly evolving sites identified specific residues in PKR that confer sensitivity or resistance to K3L. For example, an alanine to serine substitution at codon 492 reduced the ability of K3L to overcome gibbon PKR by ~100-fold in yeast assays. The finding that diversifying selection in PKR impacts its sensitivity to K3L and that K3L also appears to be rapidly evolving among poxviruses fits with an ‘arms race’ paradigm [77]. In this example, the host (PKR) and viral (K3L) genes are continually adapting to evade changes in the other, conferring a temporary advantage to the host or virus until a variant with an improved ability to antagonize the other factor arises.

These evolutionary analyses of PKR, coupled with the species-specificity of cytomegaloviruses, led us to explore whether CMVs have adapted to counteract the PKR variant in their natural hosts. The requirement for effective antagonism of PKR is illustrated by the finding that deletion of the antagonists from HCMV and MCMV completely eliminates replication in cells from their host species, but these viruses replicate as well as wild-type viruses in PKR-null cells and, in the case of MCMV, in PKR-null mice [13,23,58,68]. Similarly, rhesus CMV (RhCMV) lacking its PKR antagonist replicates ~1000-fold less efficiently than wild-type RhCMV in rhesus fibroblasts (RF). This virus does not replicate at all in HF, but it replicates efficiently in PKR-null HF [78,79]. Thus, regardless of whether CMV antagonists have “driven” adaptive changes in PKR, they have likely adapted to changes in PKR that arose due to evolutionary pressure from any number of extant or extinct viruses.

Examining the species-specific determinants of PKR antagonism by TRS1 using CMVs themselves would be a daunting task since these viruses replicate slowly, constructing mutants is cumbersome and time consuming, and their limited host range likely depends on factors in addition to PKR. Instead, we again made use of VACV∆E3L recombinants expressing TRS1 homologs from various primate CMVs (Table 1). Insertion of HCMV TRS1 into VACV∆E3L enabled efficient replication in human cells but not in Old World monkey (OWM) cells [54]. Conversely, TRS1 genes from OWM CMVs were sufficient for VACV∆E3L replication in at least some OWM cells but not in human cells [25,80]. Consistent with this work in the VACV system, insertion of TRS1 from RhCMV (rTRS1) into HCMV∆I/∆T was insufficient to support viral replication in HF [78]. These species-specific determinants of sensitivity or resistance to TRS1 suggest that CMV antagonists have indeed adapted to changes in PKR from their host species. These adaptations might constitute one barrier to cross-species transmission of CMVs.

By analyzing synthetic chimeric PKR genes, we found that mutation of a single amino acid at codon 489, which is a phenylalanine (F) in human PKR, to the serine (S) found at this position in African green monkey (Agm) PKR, rendered human PKR resistant to inhibition by TRS1 [25]. Intriguingly, this F489S mutation also improved the resistance of human PKR to the VACV K3L H47R mutant that has enhanced activity against wild-type human PKR. Residue 489 is notable as it is one of the PKR codons that has been rapidly evolving during primate evolution [44]. Moreover, based on the co-crystal structure of eIF2α and the PKR kinase domain, F489 is intimately associated with eIF2α [81]. Thus, we were surprised to find that PKR’s ability to inhibit translation tolerated almost every amino acid substitution at residue 489 [25]. However, TRS1 was only able to antagonize a small subset of the PKRs with substitutions at this site. These results are consistent with the hypothesis that the rapid evolution of PKR drives adaptation in viral antagonists, including TRS1. The ability of HCMV TRS1 to inhibit human PKR suggests that the viral gene is “ahead” in the arms race at the present time.

Using VACV∆E3L recombinants to study the species-specificity of TRS1s revealed several unexpected observations. For example, TRS1 from squirrel monkey CMV (SqmCMV), a New World monkey (NWM) virus, supports VACV∆E3L replication in cells expressing not only NWM PKR but also OWM PKR, human PKR, and even the more TRS1-resistant F489S mutant of human PKR (Table 1, [25] and unpublished data). Thus, antagonism of PKR by SqmCMV TRS1 appears to be much less constrained by species-specific determinants of PKR. A more detailed understanding of the specific interactions between the PKR and TRS1 alleles from different species may help us untangle this puzzle.

Another thus far unexplained observation is that RhCMV TRS1 (rTRS1) expressed from VACV∆E3L has very limited activity in rhesus macaque cells, although it can antagonize PKR variants in some Agm cell lines [80,82]. In contrast, rTRS1 expressed from the RhCMV genome does block rhesus PKR in RF, and it must do so for RhCMV to replicate efficiently ([78] and unpublished data). Thus, there is a discrepancy in rTRS1′s activity in the context of VACV∆E3L compared to its natural setting in RhCMV. These results again illustrate the need to confirm observations gleaned from VACV recombinants expressing foreign viral genes with analyses in the more natural homologous viral context.

**Table 1 pathogens-11-00790-t001:** Antagonism of human PKR by VACV and CMV recombinants.

Viruses		Human Cell Lines					
		HeLa	HeLa/ PKRko	HF	HF/ PKRko	HF/ rTRS1	References
**VACV**		+	+	+	+	+	[25,54,80]
VACV∆E3L (VACV∆E3L∆K3L)		−	+	−	+	−	[25,54,82]
VACV∆E3L/HCMV−TRS1		+	+	+	+		[25,55,82]
VACV∆E3L/HCMV−IRS1		+		+			[55]
VACV∆E3L/RhCMV−TRS1		−	+	−	+	+	[25,80,82]
VACV∆E3L/AgmCMV−TRS1		−	+	−	+		[25]
VACV∆E3L/SqmCMV−TRS1		+	+				[25]
**HCMV**				+	+		[23,78]
HCMV∆I/∆T				−	+		[23]
HCMV∆I/∆T/VACVE3L				+			[13]
+	Permissive to virus replication						
−	Restricted virus replication						
	Not Tested						

## 7. Role of Gene Dosage in Antagonism of PKR

The inconsistency in the anti-PKR activity of rTRS1 expressed from VACV or RhCMV might be the result of differences in the rate or levels of accumulation of dsRNA and/or rTRS1 protein between the two systems. The balance between levels of PKR and its antagonists can alter the outcome of various assays. For example, K3L rescues growth of yeast containing a single PKR allele but not of those containing two copies of the gene [48]. Additionally, VACV∆E3L containing rTRS1 does not replicate in HF, but it replicates well in HF, expressing additional rTRS1 from a transgene expressed from the host cell’s genome (Table 1, [80]). Moreover, HCMV∆I/∆T containing rTRS1 does not replicate in HF, but it does replicate in HF expressing the rTRS1 transgene [78]. This virus also replicates efficiently in HF in the presence of ISRIB, a drug that reduces the translational consequences of PKR activation [83]. Thus, successful antagonism of PKR can be sensitive to relatively small alterations in the expression levels of PKR and its antagonists.

The finding that VACV∆E3L replicates, albeit very inefficiently, in HeLa cells enabled Elde et al. to utilize an experimental evolution strategy to determine if and how VACV might adapt to antagonize human PKR more efficiently [47]. Serial passage of VACV∆E3L in HeLa cells revealed adaptations in K3L that improved its ability to antagonize human PKR and enhanced VACV∆E3L replication. In these analyses, the K3L locus consistently underwent a tandem head-to-tail amplification, resulting in overexpression of K3L and ~10-fold increase in replication. In addition, the K3L H47R mutation that showed improved resistance to human PKR in the absence of K3L amplification emerged in a subset of isolates in the experimentally evolved virus population. These results suggested a “genetic accordion model” in which a virus can rapidly adapt by gene amplification of a weak antagonist of a critical host defense factor. If an adaptive point mutation subsequently arises in one of the copies, and especially if the amplification incurs a fitness cost, the genome may subsequently collapse down to a single mutant copy.

We used a similar experimental strategy to determine how rTRS1 might adapt to overcome an inhibitory PKR [80]. To avoid selecting for K3L amplification, we inserted rTRS1 into VACV, lacking both E3L and K3L (VACV∆E∆K). The resulting virus replicated well in Agm BSC40 cells, but only weakly in an Agm fibroblast line and not at all in human or rhesus cells. Serial passage in the semi-permissive Agm fibroblasts selected for viruses containing a tandem head-to-tail amplification of the rTRS1 gene after only 3–4 passages. The adapted viruses demonstrated enhanced replication in Agm fibroblasts, and notably, these viruses also acquired the ability to replicate in human and rhesus cells, suggesting that amplification may have the potential to facilitate cross-species transmission events.

These experimental evolution results supported the conclusion that VACV is poised to adapt by gene amplification [47,80,84,85]. The fact that both the K3L and rTRS1 amplifications in the two studies became evident following just 3–4 passages suggests that VACV variants with gene amplifications may be generated often because of replication errors. If, for example, the frequency of amplification of any locus is ~10^−4^ and if the amplification confers a 10-fold replication benefit per round of replication, viruses with the amplified copy would become predominant in 3–4 passages.

During the experimental evolution of VACV∆E∆K-rTRS1, point mutants conferring improved replication arose in two other VACV genes, A24R and A35R [80,86]. Viruses with these mutations replicated well, even in the absence of rTRS1 amplification. Since A24R is a subunit of the VACV RNA polymerase, it is possible that this mutant had reduced transcriptional activity, resulting in less dsRNA and thus less requirement for a strong PKR antagonist. Very little is known about A35R, precluding predictions as to how mutations in it might rescue VACV∆E∆K-rTRS1 replication. Regardless, the appearance of mutations in genes other than the amplified antagonist that obviate the requirement of gene amplification fits with the accordion hypothesis [47].

We applied a similar experimental evolution strategy to study adaptation of rTRS1 in the context of RhCMV. Unlike HCMV and SqmCMV, which each have two genes encoding PKR antagonists, RhCMV has only one, rTRS1 (rh230, Figure 2). RhCMV replicates to a limited extent in HF but considerably better in PKR-null HF, indicating that rTRS1 is a weak antagonist of human PKR [79]. As in the VACV studies, serial passage of RhCMV in HF yielded viruses with a genomic duplication and increased expression of rTRS1. Unlike the variable number of tandem repeats detected in the VACV system, the adapted RhCMVs had only a single duplication in an inverted orientation and at a distal site. The insertion site is highly homologous to the cleavage and packaging signal found at the end of the genome, suggesting that the viral proteins that cleave the genome end during normal replication may contribute to gene duplication events. The adapted RhCMVs also had large deletions, suggesting that genome packaging constraints may result in compensatory deletions that limit the amount of DNA packaged into the capsid. VACV may tolerate larger amplifications because of the flexibility of its non-icosahedral brick-shaped nucleocapsid.

These studies reveal that both VACV and CMVs appear poised to utilize gene amplification as a means of rapidly adapting to allow an antagonist with limited activity to overcome PKR. A similar process may well occur in nature, as suggested by the presence of multiple gene families in CMVs [10]. Some of these families, which presumably arose by gene duplication followed by mutations leading to neo- or sub-functionalization of individual copies, are arranged in tandem arrays, analogous to the organization of amplified genes in the VACV experimental evolution studies. Others, including the US22 gene family, are more widely distributed around the genome, perhaps indicating that this amplification occurred earlier.

## 8. Conclusions

VACV has proven to be an invaluable tool for studying CMV-encoded antagonists of PKR. The initial identification of TRS1 and IRS1 as PKR antagonists came directly from understanding the function of E3L. The broad host range of VACV made it a highly tractable system for investigating species-specific determinants of PKR antagonism in CMVs from various hosts. Finally, the discovery of gene amplification as an adaptive mechanism in VACV guided the discovery of similar adaptive mechanisms in CMV. These experimentally driven amplification events are likely correlates of events that led to the emergence of CMV gene families over millions of years.

As might be expected based on the substantial differences between VACV and CMV, we encountered some puzzles and limitations with the application of VACV to the study of CMV genes. Both viruses are large and complex, so perhaps it is not too surprising that some properties of CMV genes may not be precisely recapitulated in the context of VACV, and vice versa. Nonetheless, our experience suggests that our understanding of VACV biology can be exploited to study the function, mechanism, and evolution of genes from CMV and other viruses.

## Figures and Tables

**Figure 2 pathogens-11-00790-f002:**
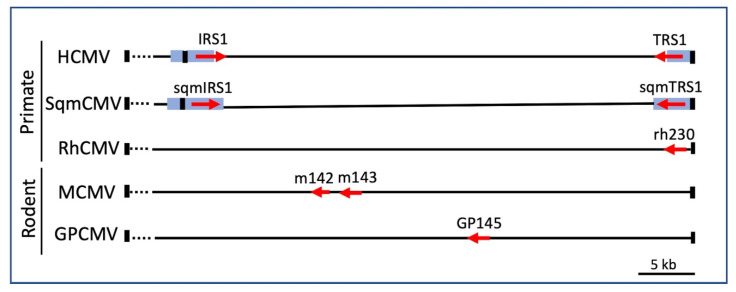
**Genomic organization of CMV PKR antagonists.** The genes encoding known PKR antagonists in the indicated CMVs, all of which are contained in the right end (~40 kb) of the genome, are depicted as red arrows (Accession numbers: HCMV, NC_006273.2; SqmCMV, NC_016448.1; RhCMV, NC_006150.1; MCMV, GU305914.1; GPCMV, NC_020231.1). HCMV IRS1 and TRS1 are partially encoded within repeats (blue boxes); thus, they have different C-termini, while the SqmCMV homologs are encoded entirely within the repeats and are identical. All CMVs have terminal repeats at each end of the genome (black rectangles), but the OWM and rodent CMVs do not have internal repeats. MCMV m142 and m143 are both required to antagonize PKR [57,58,59]. Deletion of GPCMV GP145 does not eliminate replication, suggesting that this virus may encode other not yet identified PKR antagonists [60,61].

**Figure 3 pathogens-11-00790-f003:**
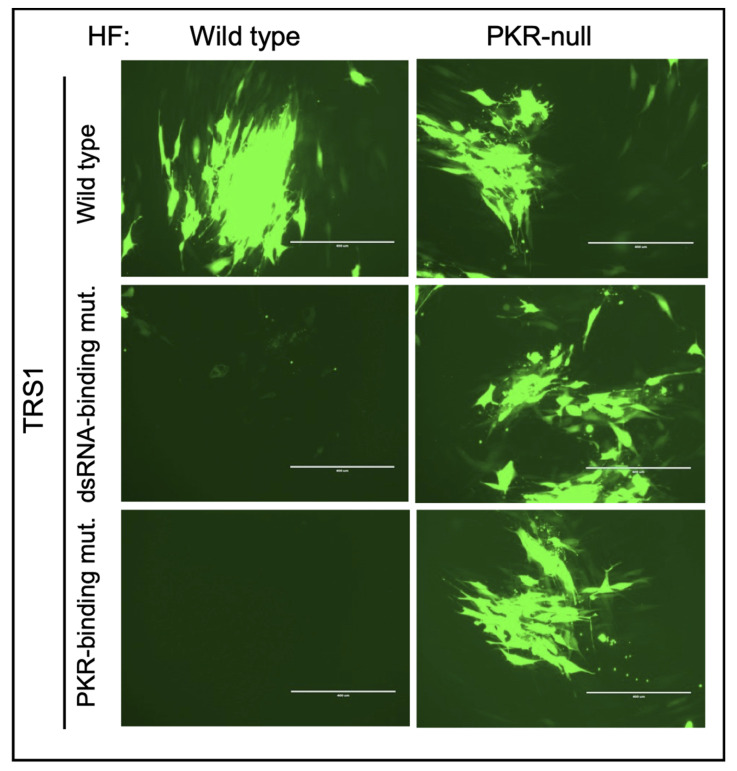
**HCMV replication depends on TRS1 binding to dsRNA and to PKR.** HF or PKR-null HF were infected with recombinant HCMVs containing EGFP and wild-type TRS1 or triple residue mutations that eliminate binding to either dsRNA or PKR. Plaques were photographed 8 days after infection (bar = 400 μm, [11,41], and unpublished data).

## Data Availability

Not applicable.

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
