# Peer review of "Antagonism of Protein Kinase R by Large DNA Viruses"

_pathogens, 2022, doi:10.3390/pathogens11070790_

Round 1
Reviewer 1 Report
Olsen et al. have reviewed the use of vaccinia virus (VACV) as a means to study inhibitors of protein kinase R in slower growing or intractable viruses, adding to the value of studying the VACV-encoded PKR inhibitors, E3L and K3L, in their natural virus and showing that VACV lacking its natural PKR inhibitors can be used to study function and evolution of other virus-encoded PKR inhibitors. The review is valuable but should be revised to make a few additional points to bring this area more into focus for interested readers.
(ln87) Authors should counterbalance there explanation of the “critical role of E3L is to counteract the PKR pathway, thereby enabling VACV replication” as this area has become more complicated and nuanced over the last 5 years as the role of E3L in suppressing Z-nucleic acid (NA) activation of ZBP1-RIPK3-MLKL necroptosis has emerged (Koehler et al., PNAS 2017; Koehler et al., Cell Host Microbe, 2021). Although the authors acknowledge the most recent work in a paragraph at ln120, the 2017 observations should also be referenced as these are key to understanding behavior of E3L mutants in vivo, supporting the idea that suppression of necroptosis seems to better explains the reason why E3L-deficient viruses are so tremendously attenuated. In any case, both of these reports are important and should be integrated early in the review. Such statements as “Deletion of both RNAseL and PKR had a larger effect, suggesting that E3L blocks both of these host defense systems in this in vivo model [28]”, cannot be as important as once throught given that the deletion of ZBP1 or RIPK3 in mice completely reverses the attenuation of E3L mutant viruses (Koehler et al., 2017). This work corrected the thinking E3L mutant behavior in mice was due solely to PKR suppression. In fact this additional function of E3L is dependent on dsRNA but independent of PKR (Koehle et al., 2021). These studies separate the role of the Z-NA binding domain of E3L that competes with ZBP1 detection of Z-NA from the dsRNA binding domain that impacts activation of PKR. In addition, very recent observations from groups such as Sid Balachandran (Zhang et al., Nature, 2022) and other recent work begins to bring the PKR and ZBP1 signaling systems into a single picture through another Z-NA binding protein, ADAR1. The direction of this type of research might also be pointed out in this timely review. Although recent findings do not address E3L function directly, they may explain observations that the authors try to explain by showing the impact of the Z-NA domain of E3L in Figure 1.
It is a bit misleading to state, “the N-terminal domain of E3L is generally not required for replication in cell culture” when it is now known that the cell lines used to study these mutants lacked the capacity for necroptosis, as pointed out clearly in the Koehler et al., work. It would be best to make a clarifying statement that leaves an appropriate impression and updates the “PKR-centric” view of E3L that has prevailed for a couple of decades.
(ln201) The ability of E3L to complement ∆IRS1/∆TRS1 HCMV raises the question of whether other complementation has been attempted using other herpesviruses, for example. Do PKR inhibitors encoded by other herpesviruses complement E3L? Does E3L complement PKR inhibition mutants in other herpesviruses? If this is a general phenomenon, it helps the authors’ general theme here. Also, is it known whether other functions of HCMV such as the IE1 gene that also counters interferon activation (Paulus and Nevels, Viruses, 2009) impacts E3L mutants or whether ∆IE1 mutants of HCMV are complemented by VACV gene function (or functions from other DNA viruses)? This area has not been reviewed and so any examples or comments would be useful to the general reader.
(ln250) As functions of MCMV are mentioned, and their abilities summarized, it might be useful to summarize more about how these are related to HCMV TRS1 and IRS1. Are they homologs?
The evolutionary aspects of the work are fascinating even if hard to rationalize, when both point mutants and amplification events seem to occur. (Why, for example, would rhesus CMV-encoded PKR inhibitor not function in rhesus cells and need “adaptation”? Why do the VACV PKR inhibitors work so broadly including in single cell eukaryotes, yeast, yet show ability to be optimized for certain species? With structures known for the VACV PKR inhibitors, can something more be said?
(ln396) Can the US22 family of HCMV genes be introduced a bit more completely as it comes up late in the review and it is not clear to the reader how this fits.
Author Response
Reviewer 1:
Olsen et al. have reviewed the use of vaccinia virus (VACV) as a means to study inhibitors of protein kinase R in slower growing or intractable viruses, adding to the value of studying the VACV-encoded PKR inhibitors, E3L and K3L, in their natural virus and showing that VACV lacking its natural PKR inhibitors can be used to study function and evolution of other virus- encoded PKR inhibitors. The review is valuable but should be revised to make a few additionalpoints to bring this area more into focus for interested readers.
(ln87) Authors should counterbalance their explanation of the “critical role of E3L is to counteract the PKR pathway,thereby enabling VACV replication” as this area has become more complicated and nuanced over the last 5 years as the role of E3L in suppressing Z-nucleic acid (NA) activation of ZBP1-RIPK3-MLKL necroptosis has emerged (Koehler et al., PNAS 2017; Koehler et al., Cell Host Microbe, 2021). Although the authors acknowledge the most recent work in aparagraph at ln120, the 2017 observations should also be referenced as these are key to understanding behavior of E3Lmutants in vivo, supporting the idea that suppression of necroptosis seems to better explains the reason why E3L-deficient viruses are so tremendously attenuated. In any case, both of these reports are important and should beintegrated early in the review. Such statements as “Deletion of both RNAseL and PKR had a larger effect, suggestingthat E3L blocks both of these host defense systems in this in vivo model [28]”, cannot be as important as once thoughtgiven that the deletion of ZBP1 or RIPK3 in mice completely reverses the attenuation of E3L mutant viruses (Koehler etal., 2017). This work corrected the thinking E3L mutant behavior in mice was due solely to PKR suppression. In fact this additional function of E3L is dependent on dsRNA but independent of PKR (Koehle et al., 2021). These studiesseparate the role of the Z-NA binding domain of E3L that competes with ZBP1 detection of Z-NA from the dsRNAbinding domain that impacts activation of PKR. In addition, very recent observations from groups such as Sid Balachandran (Zhang et al., Nature, 2022) and other recent work begins to bring the PKR and ZBP1 signaling systems into a single picture through another Z-NA binding protein, ADAR1. The direction of this type of research might also bepointed out in this timely review. Although recent findings do not address E3L function directly, they may explain observations that the authors try to explain by showing the impact of the Z-NA domain of E3L in Figure 1.
It is a bit misleading to state, “the N-terminal domain of E3L is generally not required for replication in cell culture” when it is now known that the cell lines used to study these mutants lacked the capacity for necroptosis, as pointed outclearly in the Koehler et al., work. It would be best to make a clarifying statement that leaves an appropriate impressionand updates the “PKR- centric” view of E3L that has prevailed for a couple of decades.
Response: We agree with this reviewer and the editors that the recent work on the functions of the N-terminal E3L in preventing the activation of necroptosis by Z-NA deserves more discussion. Therefore, we have added additional detail (see pages 4-5) and new relevant references as suggested (Koehler et al., 2017 and Zhang et al. 2022) as well as others (Thakur et al., 2014 and White and Jacobs 2014) which report PKR effects mediated by the E3L N-terminus.
(ln201) The ability of E3L to complement ∆IRS1/∆TRS1 HCMV raises the question of whether other complementation has been attempted using other herpesviruses, for example. Do PKR inhibitors encoded by other herpesviruses complement E3L? Does E3L complement PKR inhibition mutants in other herpesviruses? If this is a generalphenomenon, it helps the authors’ general theme here. Also, is it known whether other functions of HCMV such as the IE1 gene that also counters interferon activation (Paulus and Nevels, Viruses, 2009) impacts E3L mutants or whether∆IE1 mutants of HCMV are complemented by VACV gene function (or functions from other DNA viruses)? This areahas not been reviewed and so any examples or comments would be useful to the general reader.
Response: While VACV has been used as vector for the delivery of herpesvirus genes, for examples in vaccine and oncolytic virus development, we are not aware of published reports in which other herpesvirus genes were evaluated for rescue of VACDE3L, nor of reports attempting to complement HCMV and VACV mutants, including using the IE1 mutants as the reviewer mentioned.
(ln250) As functions of MCMV are mentioned, and their abilities summarized, it might be useful to summarize more about how these are related to HCMV TRS1 and IRS1. Are they homologs?
Response: We now mention that the MCMV gene are homologs of TRS1 and IRS1.
The evolutionary aspects of the work are fascinating even if hard to rationalize, when both point mutants andamplification events seem to occur. (Why, for example, would rhesus CMV-encoded PKR inhibitor not function in rhesus cells and need “adaptation”? Why do the VACV PKR inhibitors work so broadly including in single cell eukaryotes, yeast, yet show ability to be optimized for certain species? With structures known for the VACV PKR inhibitors, cansomething more be said?
Response: These are good questions, but we just do not know the answers yet. In this paper we note that variations between the assay systems, such as in the kinetics and abundance of expression of dsRNA, PKR, and the various antagonists might explain some of the puzzling results.
(ln396) Can the US22 family of HCMV genes be introduced a bit more completely as it comes up late in the review and it is not clear to the reader how this fits.
Response: We have added some more information earlier in the paper to explain how the US22 family of genes likely arose by amplification, following which the copies diverged to fulfill various functions. In all CMVs studied thus far, one or two of the US22 genes are PKR antagonists (top of page 8).
Reviewer 2 Report
The manuscript titled “Antagonism of Protein Kinase R by Large DNA Viruses” is a review of how VACV can be used as a valuable tool and provide a valuable guide for the study of antagonism of PKR by HCMV. The work is comprehensive in illustrating the usage of the VACV∆E3L system to identify and study PKR antagonists from several CMV species, such as HCMV TRS1 and IRS1. In addition, this review is very well organized and thus provides a useful repository of the knowledge on the topic of identification of PKR antagonists/ CMV-PKR evolutionary “arms race”/adaptation of viruses by gene amplification.
Overall, this manuscript contains excellent illustrations, which have made the review very interesting, comprehensive, and educational for the reader. It was deserving to be published in its present form.
Author Response
Reviewer 2 had no criticisms or suggestions for changes.
Reviewer 3 Report
This manuscript is well organized and thoughtfully written. PKR is a key host factor of innate immune responses, and many viruses have evolved mechanisms to evade the PKR-mediated growth inhibition. The authors began with an introduction of the two well-characterized VACV proteins that antagonize PKR actions. They then described how the VACV mutants lacking these PKR antagonists can facilitate the identification and characterization of similar antagonists encoded by other less tractable DNA viruses, focusing specifically on CMV viruses. The authors further depicted how the VACV system can be utilized to investigate the evolutionary arms race between the host PKR and the CMV antagonists, which is important for potential clinical applications. On the other hand, the authors also provided a balanced view by pointing out the caveats and limitations associated with the above approaches. I only have several minor comments:
Line 13: the word “such” should be changed to “such as”
Fig.1: The dsRNA and Z-RNA are labeled in green and purple, respectively. Since the PKR kinase domain and dsRNA binding domains are also shown in the same colors, it may help communicate the information in the figure if the dsRNA and Z-RNA are depicted in different, more contrasting colors.
Fig.1 legend: In addition to ref #12, the authors may consider to also cite the paper by Mayo et al. (Biochemistry 2019) which provides an additional insight into PKR activation.
Author Response
Reviewer 3: I only have several minor comments:
Line 13: the word “such” should be changed to “such as”
Response: We made that correction.
Fig.1: The dsRNA and Z-RNA are labeled in green and purple, respectively. Since the PKR kinase domain and dsRNA binding domains are also shown in the same colors, it may help communicate the information in the figure if the dsRNAand Z-RNA are depicted in different, more contrasting colors.
Response: We have revised the coloring in Fig. 1 in accordance with the reviewer’s suggestion.
Fig.1 legend: In addition to ref #12, the authors may consider to also cite the paper by Mayo et al.
(Biochemistry 2019) which provides an additional insight into PKR activation.
Response: We have added that reference as suggested.